# Evidence for Using ACQUIRE Therapy in the Clinical Application of Intensive Therapy: A Framework to Guide Therapeutic Interactions

**DOI:** 10.3390/bs13060484

**Published:** 2023-06-07

**Authors:** Stephanie C. DeLuca, Mary Rebekah Trucks, Dorian Wallace, Sharon Landesman Ramey

**Affiliations:** 1The Fralin Biomedical Research Institute’s Neuromotor Clinic, Roanoke, VA 24016, USA; mrebekah@vtc.vt.edu (M.R.T.); wdorian6@vtc.vt.edu (D.W.); slramey@vt.edu (S.L.R.); 2The School of Neuroscience, Virginia Tech, Blacksburg, VA 24061, USA; 3The Department of Pediatrics, Virginia Tech Carilion School of Medicine, Roanoke, VA 24016, USA

**Keywords:** cerebral palsy, traumatic brain injury, hemispherectomy, hemiparesis, quadriparesis, intensive therapy, ACQUIRE therapy, pediatric constraint-induced movement therapy, hand arm bimanual therapy

## Abstract

Intensive therapies have become increasingly popular for children with hemiparesis in the last two decades and are specifically recommended because of high levels of scientific evidence associated with them, including multiple randomized controlled trials and systematic reviews. Common features of most intensive therapies that have documented efficacy include: high dosages of therapy hours; active engagement of the child; individualized goal-directed activities; and the systematic application of operant conditioning techniques to elicit and progress skills with an emphasis on success-oriented play. However, the scientific protocols have not resulted in guiding principles designed to aid clinicians with understanding the complexity of applying these principles to a heterogeneous clinical population, nor have we gathered sufficient clinical data using intensive therapies to justify their widespread clinical use beyond hemiparesis. We define a framework for describing moment-by-moment therapeutic interactions that we have used to train therapists across multiple clinical trials in implementing intensive therapy protocols. We also document outcomes from the use of this framework during intensive therapies provided clinically to children (7 months–20 years) from a wide array of diagnoses that present with motor impairments, including hemiparesis and quadriparesis. Results indicate that children from a wide array of diagnostic categories demonstrated functional improvements.

## 1. Introduction

Historically, pediatric rehabilitation has been eclectic in therapy delivery models in large part because of the need for individualized services in clinical models of care where the treated children have a wide range of diagnoses as well as variations in functional abilities and severity levels associated with their disability [1]. In this regard, the past few decades in pediatric rehabilitation have seen the development of many evidence-based therapeutic approaches based on five common constructs [2,3,4,5,6,7,8,9,10]:Treatment delivery via an intensive therapeutic burst (i.e., many hours each day on multiple consecutive days per week across multiple weeks);Goal-directed activities with componential parts of therapy activities progressing toward increased movement, function, and skill;Active engagement of the child’s current sensory-motor skills throughout all therapy sessions;Activity selection guided by the child’s interest with therapeutic modification to accomplish movement, function, and skill goals during playful, success-oriented interactions; andUse of operant conditioning techniques, where positive reinforcement is provided to teach skills via variations in the contingencies of reinforcement to successively shape the child’s skills toward a targeted goal.

The most documented of these constructs has been the concept of therapy being delivered in high dosages or in intensive therapeutic bursts of treatment where many hours of therapy are administered multiple days a week (often daily) within a time period that is limited to a few weeks. Two of the most well-known high-dosage therapies were designed specifically for hemiparetic cerebral palsy [2,3,4,5,6,7,8,9,10]. Pediatric constraint-induced movement therapy (PCIMT) and hand arm bimanual therapy (HABIT) are now routinely recommended as treatment approaches for children with hemiparesis to improve motor and functional skills [5,7,8,10,11]. Recent guidelines even recommend that these approaches begin in infancy [11] despite the fact that there is substantial evidence of the use in infancy being more limited [12]. These intensive high-dosage therapies have been the subject of numerous clinical trials and numerous systematic reviews serving as the basis for these recommendations, and high-dosage intensive therapies are now becoming commercially available. Despite these advances, these approaches are far from the clinical norm as the standard of care for children with disabilities, even for children with the specified diagnosis of hemiparesis. Rather, they are limitedly available for families that specifically seek these services.

There are many reasons for the limited dissemination of these therapeutic approaches and why they are not considered standard of care. These include limited coverage by third party payers for such services and the ability of the healthcare system to adequately provide these services within the current institutionalized models of care. However, there is another issue that is rarely addressed. Are therapists adequately prepared and trained to deliver intensive models of therapy? Intensive therapy models and distributed practice models both have a goal of increasing motor skills, but almost by definition they must approach the process of learning and teaching motor skills differently. Pediatric therapists who see children once or twice a week, usually for under an hour, must quickly identify a limited focus area at each visit and primarily educate parents and caregivers to focus on that one area to promote learning. Next, with multiple days between visits, they must rely on parental reports to understand reactions, levels of learning that promote gains or losses in motor skills, and then once again quickly decide on either the same or different focus area for the new visit. This model provides difficult decision points for therapists when children rarely have a singular need.

In contrast, most, if not all, high-dosage intensive therapies that have high-quality evidence to support their use stem from scientific investigations that were built, at least in part, on learning theories [13,14,15]. Protocols were built to include sufficient time where direct observations of child responses and reaction to those responses could be implemented across multiple repetitions. Furthermore, these protocols were built on the concept that the promotion of motor skills occurred within multi-contextual developmental domains that were interacting, and that those interactions were also a reflection of complex neurological pathways. This concept means that skill development is built across domains simultaneously (e.g., motor skills depend to a certain degree on cognitive skills and vice versa) and all of them need to be considered in the learning process, once again requiring time for a therapist to consider these cross-domain impacts. Above all, these scientifically investigated protocols were built on learning principles to guide therapeutic decision-making.

For example, decades of learning the literature has detailed the variations in reinforcement schedules needed to promote learning across differing ages [16,17]. A reinforcement that is delayed by a second for an infant negates the learning potential in that moment, whereas for a child of 3 or 4 years of age, the schedule of reinforcement has a broader time span to promote the desired learning [18,19,20]. Similarly, the scaffolding or progression of skill in operant conditioning must be progressed at certain levels of proficiency. Progression before 70–80% proficiency at a given level or even waiting until a child is completely proficient at a given level can stop or alter the progression of learning. These concepts were built on observations of children who were typically developing, but they have been robust in the promotion of learning across diagnostic categories and learning styles [13,14,16,21,22,23,24]. They were a primary and integral part of the early protocols and scientific investigations into intensive therapies. However, unlike the concept of high-dosage, they have been less built into the therapeutic lexicon and dissemination of high-dosage intensive therapies. They are also not routinely taught via therapy curriculums, causing many therapists to be ill-equipped at providing a high-dosage therapy clinically that maintains the levels of efficacy seen in clinical trials.

We sought to address this as we began to try to disseminate our research protocols and use of intensive therapies. The ACQUIRE framework as seen in Figure 1 represents a complex and reciprocal interplay between the child and the therapist that is under constant evolution because of the many different variables impacting therapeutic interactions. It was designed to inform and assist therapists in the delivery of high-dosage intensive therapy in order to create high-quality densely packed therapy activities that involve needed repetitions and skill refinement to promote motor learning. 

At the heart of the framework is a cyclical set of steps based on operant conditioning. We termed the central operant conditioning process as the MR3 Cycle: movement, reinforcement, repetition, and refinement [14]. This pattern guides the progression of learning by scaffolding supports and demands toward a targeted motoric and functional outcome or learning goal. As stated above, a key component of this process is to allow the sufficient repetition of tasks (via massed practice) to promote proficiency, combined with an understanding of when and how tasks might be refined and progressed by providing appropriate types, levels, and schedules of reinforcement. Refinement and progression are key. Massed practice alone does not move learning toward a target. The model also seeks to define the therapeutic environment in a manner to help the therapist understand the many components that overtly and subtly impact learning and the progression of skill. For example, a request for a movement that is above a child’s skill level may result in a failed attempt at a movement, or it may also result in the child not responding at all. In both instances, a therapist must evaluate and react to the child and the demands of the task appropriately to promote learning. A parent entering the room may distract a child from a movement attempt where they were previously successful, making them unsuccessful secondary to the distraction. The therapist must recognize the basis of this failed attempt and respond to re-direct the child’s attention. The process is quite complex. The collective and individualized attention, awareness, perception, and understanding of the task for both the therapist and the child are almost in a constant state of flux, creating unique demands on the therapy process; demands which therapists must be prepared to guide. 

Figure 2 shows the complex decision-making process involved in that guidance. The process starts with a choice of an appropriate task. Remember that all tasks are dependent on multiple developmental domains, and thus many complexities must be considered. The choice of task must be motivating and meet a child’s current ability levels. For example, if a parent has a goal that a child uses a paretic arm in dressing, but that child does not yet reach with their paretic arm, a choice for a task might be only to reach forward with the paretic arm toward a motivating toy. At first, that reach may even be untargeted. The therapist requests the child to reach for the toy with playful engagement, while providing cues and instructions to the child. The cues and instructions need to be specific and should include a modeling of the task. After modeling, if the child’s attempt is not successful, the therapist may include hand-over-hand facilitation to help the child to complete the task in order to reinforce their attempts and allow a feeling of success. With each task request, the therapist must then allow sufficient time for a child to respond. This is key within a therapeutic context because not only does a task stem from multiple developmental domains, but a child’s limitations may also be linked to many developmental domains (e.g., planning and processing limitations). Then, as shown in Figure 2, there are three possible child responses: a child successfully completes the requested task, the child is unsuccessful at completing the task, or the child does not respond. The MR3 operant conditioning cycle dictates that the therapist must respond, but the response is dependent on many constructs that a therapist needs to immediately evaluate. In the above example, if the child reaches forward and performs this on a sufficient number of occasions, the therapist may progress the skill by adding a reaching target (e.g., a large lever on a toy). As progression continues across hundreds of repetitions of reaching, the therapist may increase the complexity of having an open hand or to reach in different directional planes. Demands of a task can even be increased by changing how the request is made. A therapist may proceed from pairing a verbal and tactile request to merely a verbal request. All of this is relatively child-dependent and context-dependent because at each point in the process, the therapist must consider the many components impacting the child’s learning. The ACQUIRE framework is meant to provide organization for many of the constructs to be considered.

We have now used this therapeutic process to train many therapists across two therapeutic research clinics, multiple clinical trials, and in the training of doctoral candidates in therapy professions. The clinical trials primarily focused on training therapists to complete intensive therapies where dosage levels and differing constraint types were used, including a bimanual approach involving no constraint. In our research clinics, we collected data gathered as practice-based evidence using the ACQUIRE therapy model while providing high-dosage clinical services across many different diagnostic categories, thus addressing another need in the dissemination of high-dosage intensive therapy models and their use across heterogenic clinical populations. We present that data below.

## 2. Materials and Methods

### 2.1. Participants/Clients

The sample is a convenience clinical sample collected across two research clinics at two different universities. The clinics were not operated concurrently. Children were brought to each of the clinics by caregivers to specifically receive intensive therapy services based on the ACQUIRE model of therapy. Families often sought services after interaction with other families via online support services. Children ranged in diagnoses, but were pre-screened by clinic personnel to ensure that there were no concerns about the children participating in intensive therapy. All levels of severity were included as long as the child was deemed medically stable, meaning that there were no existing movement/range of motion or behavioral requirements necessary for a child to receive services. While some children were quadriparetic, all children had some level of asymmetric functioning. Ethical approvals were obtained for all data collection at both clinics. All participants signed informed consent forms, permitting data during clinic-based services to be collected, analyzed, and published. The sample presented represents a subset of all clinical data collected at the two clinics. Some clinical data were unavailable for analyses because they have not yet been entered into the current database. The clinics did provide some children multiple epochs of intensive therapy if requested, but data from children who received additional epochs of treatment were not included. Children were also excluded if they did not finish the course of planned intervention (e.g., because of illness).

### 2.2. Intervention

ACQUIRE therapy was delivered between 4–6 h each weekday for 4 consecutive weeks, creating intensive therapy epochs of 80–120 therapy hours. Variations in dosage occurred for a variety of reasons, but most often was a function of parental requests or limitations in clinical coverage. A full arm constraint was used for children with hemiparesis via a PCIMT model called ACQUIREc therapy [13,14]. ACQUIREc Therapy is a specific form of ACQUIRE Therapy. Both the clinics were initially started to deliver this manualized PCIMT approach that served as the basis for our further development of the ACQUIRE framework. The basis for therapy for children with quadriparesis was ACQUIRE therapy, in which a constraint may or may not have been used, depending on the child’s motor involvement, levels of asymmetry, and individual goals of the child and family.

### 2.3. Assessments

A battery of qualitative and clinical (quantitative) assessments were completed within 2 days prior to and after completion of ACQUIRE therapy. Quantitative data presented in this paper include the Emerging Behaviors Scale (EBS) [14], the Assisting Hand Assessment (AHA) [26], and the Pediatric Motor Activity log (PMAL) [14]. All assessments were primarily designed to examine asymmetric functioning of the upper extremities. The EBS is a count of 30 possible arm and hand skills. The AHA is a measure designed to examine bilateral performance of an assisting hand, and the PMAL is a parental report measure of 22 arm and hand skills where the parent reports how well and how often the child uses the more paretic arm and hand across a 5-point Likert scale, where 0 indicates no ability and no use of the more paretic arm and hand items where parents rate the functioning of the more paretic arm and hand across two scales that range between 0–5. The ‘how often’ scale provides an ordinal level ranking of how frequently children use the more paretic arm and hand and the ‘how well’ scale provides an ordinal level ranking of the quality of skills of the more paretic arm and hand.

### 2.4. Data Analysis

Descriptive statistics were prepared for sample characteristics and all quantitative data. Data between pre- and post-treatment were examined with repeated measures analysis of variance (ANOVA). Change scores were generated to compare between diagnostic categories and to compare outcomes between children with hemi- and quadriparesis. Analyses fused counts of the 30 potential behaviors for the EBS, logit scores for the AHA, and averages for PMAL ‘how often’ and ‘how well’ scales. Descriptive data were reported by parents.

## 3. Results

### 3.1. Participants/Clients

Participants/clients data are based on 139 children between 7 months to 20 years of age (mean = 62.1, S.D. = 53.41). There were 61 females and 78 males representing 44 and 56% of the sample, respectively. Ethnic or racial categories were not routinely recorded by the clinic, but retrospective examination of data indicated that about 10% of the sample represented children from racial and ethnic categories other than white or Caucasian. Table 1 presents the numbers of children by diagnostic categories compared across those with hemiparesis versus quadriparesis. ACQUIRE therapy was delivered for 6 h each weekday for 4 consecutive weeks for 118 of these children, making a dosage of 120 h of therapy. For four children, scheduling issues with their families caused three weeks to be delivered instead of four weeks at 6 h per day, resulting in a total dosage of 90 h. Four of twelve children’s parents and therapists collaboratively decided to complete 4 h of therapy 5 days a week for 4 weeks, resulting in a total dosage of 80 h.

### 3.2. The Emerging Behavior Scale

The EBS was the most consistent measure used across all children, and analysis included n = 121. Across all diagnostic categories and paresis types, children gained an average of 9.15 (S.D. = 5.98) new behaviors. Repeated measures ANOVA indicated a main effect of time between pre- to post-treatment with F = 23.51, *p* < 0.001. There were no significant differences found based on diagnosis or type of paresis. Table 2 shows mean change scores by diagnostic category and paresis type. Across all children with hemiparesis, the mean = 9.36 (S.D. = 6.03; n = 106), and across all children with quadriparesis, mean = 7.67 (S.D. = 5.51, n = 15). Results suggest that children with a variety of diagnoses that present with either hemi- or quadriparesis gained new unilateral skills.

### 3.3. The Pediatric Motor Activity Log

The PMAL was completed by 97 parents, of which 87 of the children presented with hemiparesis and 10 children presented with quadriparesis. Across all diagnostic categories and paresis types, parents rated their children as having increased amounts of use for the more paretic arm with a mean change = 2.17 (S.D. = 1.07), and they rated that their children’s abilities with their more paretic arm and hands increased with a mean change = 1.54 (S.D. = 0.94). Repeated measures ANOVA, considering both the ‘how often’ and ‘how well’ scales indicated significant main effects of time between pre- to post-treatment. The ‘how often’ scale produced an F = 32.74, *p* < 0.001, and the ‘how well’ scale F = 17.58, *p* < 0.001. There were no significant differences found based on diagnosis or type of paresis. Table 2 shows mean changes for each scale by diagnostic category and paresis type. Across all children with hemiparesis, the mean = 2.03 (S.D. = 1.07; n = 87), and across all children with quadriparesis, mean = 1.91 (S.D. = 1.1, n = 10). Results demonstrate that parents reported changes in both the quality of their children’s skills with the more paretic arm and hand, but reported even more changes in how frequently their children were using their more paretic arm and hand immediately after intensive therapy. Notably, this finding is true across diagnostic and paresis types (i.e., hemi- or quadriparetic). 

### 3.4. The Assisting Hand Assessment

Data from the AHA were available for 26 children. All children were diagnosed with hemiparetic CP (n = 25) or stroke (n = 1). Three children were quadriparetic but highly asymmetrical. Across all diagnostic categories and paresis types, children gained an average of 11.19 (S.D. = 7.55) logit score points. Repeated measures ANOVA indicated no significant main effect of time between pre- to post-treatment, only a trend toward significance with F = 3.98, *p* = 0.058. There were no significant differences found based on diagnosis or type of paresis. Figure 3 demonstrates the pre- to post-changes by paresis type. Despite the fact that there was only a trend toward significance in this measure, as Figure 3 demonstrates, there were positive changes in all children. The lack of significance is likely related to the heterogeneity of the children and the fact that the measure was designed for measuring children with hemiparesis. As stated above, the children with quadriparesis that we chose to use the measure on in this clinical sample were extremely asymmetric. Results suggest that children with a variety of diagnoses that present with either hemi- or quadriparesis gained abilities to use their more paretic arm and hand in bimanual activities. 

## 4. Discussion

The usability of high-dosage intensive therapies in clinical settings is impacted by multiple issues. Two primary issues not adequately considered are, firstly, that clinical populations are usually highly heterogeneous, and secondly, that therapists are not adequately prepared to implement these high dosage therapeutic approaches. We sought to address both of these issues in this paper.

The ACQUIRE framework provides a more defined and detailed representation of the interactive therapeutic processes and provides specific constructs that impact learning during high-dosage intensive therapy. These concepts have a strong foundation in theories about learning, specifically operant conditioning, and should be seen as additive components to high-dosage therapies which have traditionally been defined more fundamentally in terms of frequency, intensity, and timing of therapy. Therapeutic encounters are complex, and the ACQUIRE framework and associated decision-making process seek to make therapeutic learning a collaborative and more definable interaction between the therapist and the child and must include sufficient time and opportunity for massed but refined processes to promote learning. In order to use high-dosage intensive therapies, therapists must be trained to understand these constructs. Concomitantly, high-dosage therapies and these processes allow therapists ample time to consider the whole child who is intermixing developmental domains almost on a continuous basis throughout development. We theorize that limited time constraints do not allow therapists to consider the interplay from both the therapeutic delivery side and the internal development side of the child, and that this is a major impediment for therapy delivery that routinely and systematically promotes the development of skills.

We have now used the ACQUIRE framework to train dozens of therapists and guide intensive therapy epochs for hundreds of children across many diagnoses. The data across etiologies presented in this paper begin to address the question about the use of high-dosage intensive therapy beyond merely hemiparesis. Across six diagnostic categories that included CP, CVA, TBI, AVM, hemispherectomy, and others (e.g., microcephaly, tumor resections), children consistently responded positively to receiving high-dosage intensive ACQUIRE therapy and gained movements and functional skills. The data in this paper were primarily focused on developing and measuring motor skills, primarily that of the upper extremity in children with multiple types of diagnoses. Importantly, children with varied diagnoses improved. In addition, we collapsed across levels of paralysis and comparing children with hemiparesis to children with quadriparesis, and both groups improved on the number of skills developed (e.g., the EBS) and by parental report (e.g., the PMAL). Magnitudes of change favored children with hemiparesis but all responded positively. 

There are limitations to the data we present. First and foremost, we present data collected for clinical purposes and to internally understand if the intensive therapy services we were providing were indeed producing positive changes. Both of the clinics were initially designed to provide our manualized version of pediatric constraint-induced movement therapy, ACQUIRE Therapy [14], and we have published numerous clinical trials [26,27,28,29] based on this protocol, but our assessment and expanded treatment protocols were built on this legacy by incorporating all the components of that protocol outside the constraint with children who had bilateral paresis. Our routinely used measurements of change are limited for this reason. We often saw changes and parents reported changes in other motor areas (e.g., gross motor skills) and developmental domains (e.g., language) not routinely tested. We further recognize that children in this sample received a minimum of 80 h of therapy within a four-week period. While this meets recommendations made within the literature [3], it is well above the current standard of practice and perhaps more importantly what is routinely covered by third party reimbursement. This is a major factor that prevents many children from receiving intensive therapy. While our data cannot adequately address the health disparities associated with this fact, it is incumbent upon us to recognize it as a limitation.

## 5. Conclusions

The field of pediatric rehabilitation, or at least therapists on a provider level, appear to be increasing the amount of therapeutic services provided to children in traditional settings [25,26,27,28,29]. Perhaps this change, in part, is due to the overall increase in pediatric therapy services, driven mostly by demand of caregivers. There also appears to be an increase in the number of facilities providing intensive pediatric therapy services, again likely driven by caregiver demand. While the increase in intensity is necessary, our experience with providing intensive services for children with neuromotor impairments, both in research and clinic settings, has led us to a greater understanding of the multiple necessary components that exist for the effective and efficacious delivery of high-quality intensive therapies that excel beyond merely the component of dosage. Dosage is a highly recognized and needed component [3,25], but it is not the only essential component for intensive therapies to be delivered in a manner that maximizes efficacy. The ACQUIRE framework and therapy delivery are meant to guide the necessary interactions between the therapist and the child in order maximize intensive therapy services for both the therapeutic delivery side provided by the therapist and on the developmental side for the child. As the field makes further investments into pediatric rehabilitation to utilize high-dosage therapies, we need to better establish how we prepare therapists to deliver these therapeutic efforts. 

The next steps are always difficult when considering how to translate research findings into practice. Defining these intensive therapies in terms of frequency, density, and dosage was a major addition to the pediatric rehabilitation field driven by research, but it has led to an eclectic mix of therapies with varying levels of results. The next steps include a greater standardization of guiding methodologies and decision-making processes used to deliver intensive therapies. 

## Figures and Tables

**Figure 1 behavsci-13-00484-f001:**
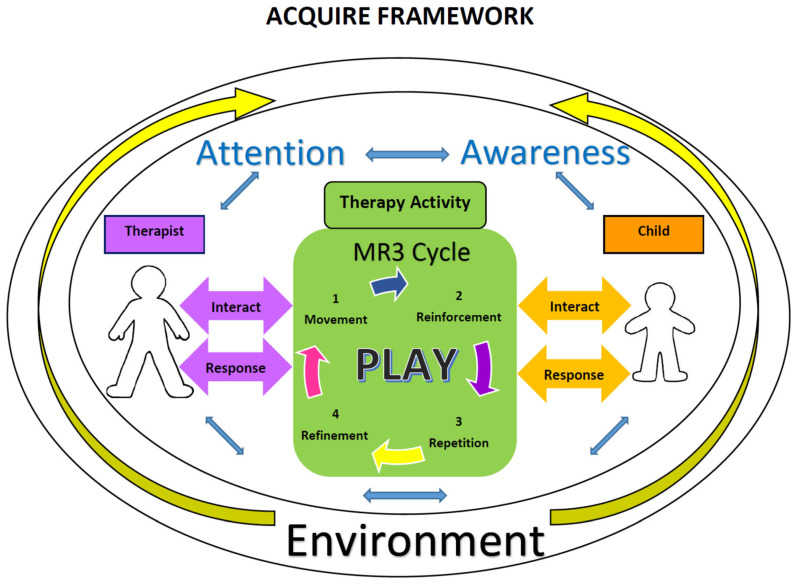
The ACQUIRE framework [8,13,14,25].

**Figure 2 behavsci-13-00484-f002:**
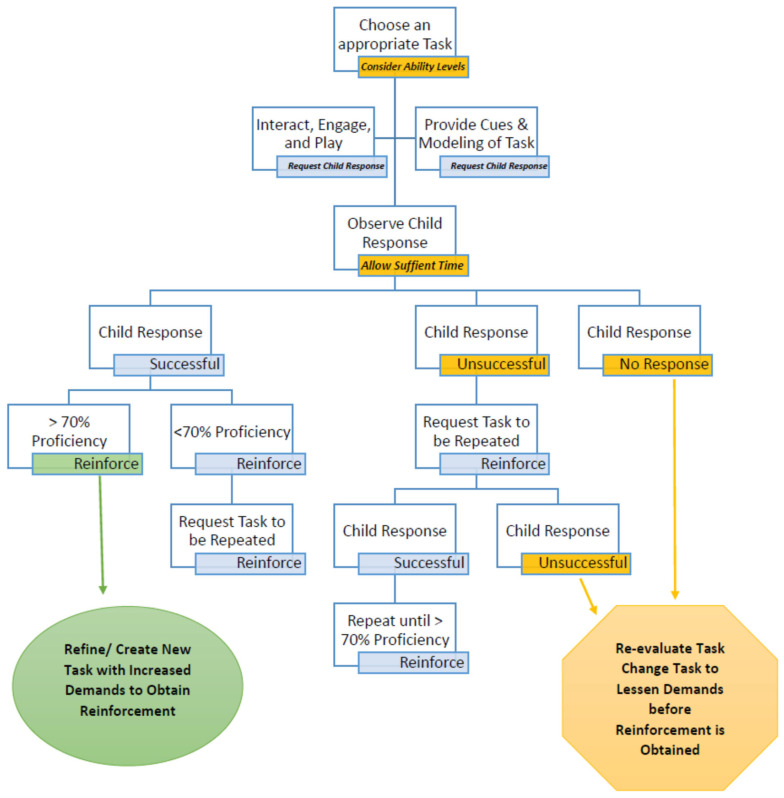
Therapeutic activity decision tree.

**Figure 3 behavsci-13-00484-f003:**
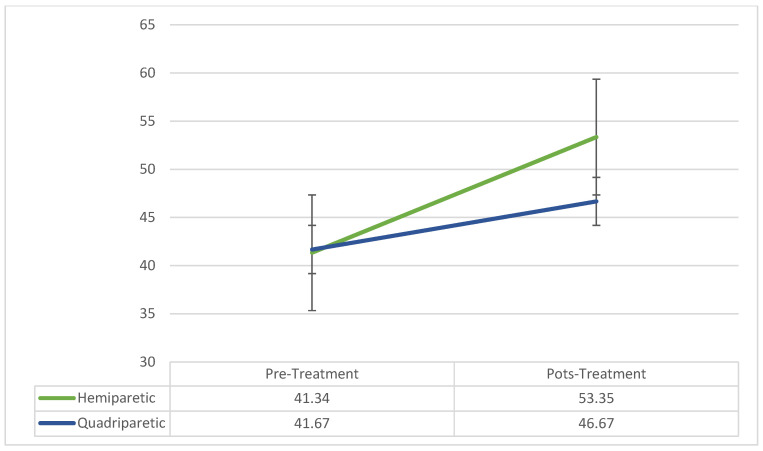
Pre- to post-treatment AHA logit scores.

**Table 1 behavsci-13-00484-t001:** Diagnostic categories by type of paresis.

	Hemiparesis	Quadriparesis
Cerebral Palsy (CP)	74	10
Cerebral Vascular Accident (CVA)	25	3
Arteriovenous Malformation (AVM)	1	0
Traumatic Brain Injury (TBI)	9	1
Hemispherectomy	4	0
Not Otherwise Specified Motor Delay	8	4
	121	18

**Table 2 behavsci-13-00484-t002:** Mean Gain Scores (S.D.) by diagnostic categories.

Diagnosis		Emerging Behaviors Scale		Pediatric Motor Activity Log			
				Frequency of Use		Quality of Movement	
CP	Hemiparesis	9.71 (4.96)	n = 69	2.04 (1.14)	n = 45	1.50 (0.99)	n = 45
	Quadriparesis	7.56 (5.62)	n = 9	1.73 (0.68)	n = 6	1.47 (0.84)	n = 6
CVA	Hemiparesis	9.5 (9.29)	n = 20	2.18 (1.11)	n = 22	1.76 (1.05)	n = 22
	Quadriparesis	11.67 (6.43)	n = 3	2.39 (1.94)	n = 3	1.11 (0.96)	n = 3
TBI	Hemiparesis	7.86 (6.62)	n = 7	1.93 (0.97)	n = 8	1.55 (0.90)	n = 8
	Quadriparesis	5.00	n = 1	1.46	n = 1	1.42	n = 1
Hemispherectomy	Hemiparesis	9.00 (6.56)	n = 3	1.56 (0.91)	n = 4	2.00 (0.69)	n = 4
	Quadriparesis	N/A	n = 0	N/A	n = 0	N/A	n = 0
AVM	Hemiparesis	5.00	n = 1	2.63	n = 1	1.23	n = 1
	Quadriparesis	N/A	n = 0	N/A	n = 0	N/A	n = 0
Other	Hemiparesis	7.5 (1.80)	n = 6	1.78 (0.94)	n = 1	1.14 (0.71)	n = 1
	Quadriparesis	3.5 (0.71)	n = 2	N/A	n = 0	N/A	n = 0
			N = 121		N = 97		N = 97

## Data Availability

Data collected were collected for clinical purposes and therefore individual data are not available for distribution or public use.

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
