# Peer review of "Evidence for Using ACQUIRE Therapy in the Clinical Application of Intensive Therapy: A Framework to Guide Therapeutic Interactions"

_behavsci, 2023, doi:10.3390/bs13060484_

Round 1

Reviewer 1 Report

clean well written study

it is beneficial to expand study to additional conditions

this type of therapy has been shown to be effective. What are the next steps in the process of adopting this type of therapy as standard practice

this study would be improved with a greater analysis of limitations and next steps

Author Response

Reviewer 1 Comments:

clean well written study

it is beneficial to expand study to additional conditions

this type of therapy has been shown to be effective. What are the next steps in the process of adopting this type of therapy as standard practice

we have talked more about this in discussion section.

this study would be improved with a greater analysis of limitations and next steps

We thank the reviewer for the positive comments, many of which matched other reviewers.

Per the request made we have expanded our limitations & conclusion sections, with conclusions now discussing next steps.

Reviewer 2 Report

Thank you very much for the opportunity to review this research. I think that the research could be improved with very few changes, which I will suggest below.

Abstract

Select better keywords

Introduction

I write a series of references that can improve this section

Baker, A., Niles, N., Kysh, L., & Sargent, B. (2022). Effect of Motor Intervention for Infants and Toddlers With Cerebral Palsy: A Systematic Review and Meta-analysis. Pediatr. Phys. Ther., 34(3), 297–307.

Eliasson, A. C., Krumlinde-Sundholm, L., Gordon, A. M., Feys, H., Klingels, K., Aarts, P. B. M., Rameckers, E., Autti-Rämö, I., & Hoare, B. (2014). Guidelines for future research in constraint-induced movement therapy for children with unilateral cerebral palsy: an expert consensus. Dev. Med. Child Neurol., 56(2), 125–137

Hoare, B., & Eliasson, A. C. (2014). Evidence to practice commentary: Upper limb constraint in infants: Important perspectives on measurement and the potential for activity-dependent withdrawal of corticospinal projections. In Phys. Occup. Ther. Pediatr. (Vol. 34, Issue 1, pp. 22–25). Phys Occup Ther Pediatr.

Results

I think it would be convenient to mark the total number of hours, the possible relationship of the total number of hours with the variables-results

During the text it is suggested that there is improvement with this program, but it should be noted that there is clinical but not statistical improvement, (this may be a limitation)

Discussion

Talk about the total number of hours and whether it is important to have a high or low dose, when it is known that the total number of hours matters

Regards

Author Response

Reviewer 2 Comments:

Thank you very much for the opportunity to review this research. I think that the research could be improved with very few changes, which I will suggest below.

We than the reviewer for their positive comments and suggested changes. We respond to each change below. 

Abstract

Select better keywords

We have added Pediatric Constraint-Induced Movement Therapy, & Hand Arm Bimanual Therapy, we interpreted the suggestion as needed additions.

 Introduction

I write a series of references that can improve this section

Baker, A., Niles, N., Kysh, L., & Sargent, B. (2022). Effect of Motor Intervention for Infants and Toddlers With Cerebral Palsy: A Systematic Review and Meta-analysis. Pediatr. Phys. Ther., 34(3), 297–307.

Eliasson, A. C., Krumlinde-Sundholm, L., Gordon, A. M., Feys, H., Klingels, K., Aarts, P. B. M., Rameckers, E., Autti-Rämö, I., & Hoare, B. (2014). Guidelines for future research in constraint-induced movement therapy for children with unilateral cerebral palsy: an expert consensus. Dev. Med. Child Neurol., 56(2), 125–137

 Hoare, B., & Eliasson, A. C. (2014). Evidence to practice commentary: Upper limb constraint in infants: Important perspectives on measurement and the potential for activity-dependent withdrawal of corticospinal projections. In Phys. Occup. Ther. Pediatr. (Vol. 34, Issue 1, pp. 22–25). Phys Occup Ther Pediatr.

 We have added these reference and an additional comment about the limitations about the evidence of intensive therapies in infants.

Results

I think it would be convenient to mark the total number of hours, the possible relationship of the total number of hours with the variables-results

This is now added.

During the text it is suggested that there is improvement with this program, but it should be noted that there is clinical but not statistical improvement, (this may be a limitation)

We believe the reviewer was referring to the Assisting Hand Assessment Data in this comment. We have more clearly point this out within the text.

Discussion

Talk about the total number of hours and whether it is important to have a high or low dose, when it is known that the total number of hours matters

We have added some additional comments within the limitation section.

Reviewer 3 Report

Excellent paper! I appreciated your expertise in this area of pediatric rehabilitation. The paper is well structured and has a solid theoretical and practical base. I suggest a review, maybe an extension, of the bibliography.

Otherwise, I see no need for changes

Author Response

Reviewer 3 Comments:

Excellent paper! I appreciated your expertise in this area of pediatric rehabilitation. The paper is well structured and has a solid theoretical and practical base. I suggest a review, maybe an extension, of the bibliography.

Additional references were also suggested by an additional reviewer we have added the following references:

Baker, A., Niles, N., Kysh, L., & Sargent, B. (2022). Effect of Motor Intervention for Infants and Toddlers With Cerebral Palsy: A Systematic Review and Meta-analysis. Pediatr. Phys. Ther., 34(3), 297–307.

Eliasson, A. C., Krumlinde-Sundholm, L., Gordon, A. M., Feys, H., Klingels, K., Aarts, P. B. M., Rameckers, E., Autti-Rämö, I., & Hoare, B. (2014). Guidelines for future research in constraint-induced movement therapy for children with unilateral cerebral palsy: an expert consensus. Dev. Med. Child Neurol., 56(2), 125–137

 Hoare, B., & Eliasson, A. C. (2014). Evidence to practice commentary: Upper limb constraint in infants: Important perspectives on measurement and the potential for activity-dependent withdrawal of corticospinal projections. In Phys. Occup. Ther. Pediatr. (Vol. 34, Issue 1, pp. 22–25). Phys Occup Ther Pediatr.

Otherwise, I see no need for changes.

We thank the reviewer for the positive comments!